# Impact of PCV13 and PPSV23 Vaccination on Invasive Pneumococcal Disease in Adults with Treated Rheumatoid Arthritis: A Population-Based Study

**DOI:** 10.3390/microorganisms12102073

**Published:** 2024-10-16

**Authors:** Carlos A. Alvarez, Ronald G. Hall, Suzy Lin, Aaron R. Perkins, Eric M. Mortensen

**Affiliations:** 1Jerry H. Hodge School of Pharmacy, Texas Tech University Health Sciences Center, Dallas, TX 75235, USA; ronald.hall@ttuhsc.edu (R.G.H.2nd); suzy.w.lin@outlook.com (S.L.); aaron.perkins@ttuhsc.edu (A.R.P.); 2Center of Excellence in Real World Evidence, Dallas, TX 75235, USA; 3VA North Texas Health Sciences Center, Dallas, TX 75216, USA; 4UCONN School of Medicine, Farmington, CT 06030, USA; mortensen@uchc.edu

**Keywords:** pneumococcal vaccine, rheumatoid arthritis, immunocompromised, PCV13, PPSV23, invasive pneumococcal disease

## Abstract

On-time receipt of pneumococcal vaccines is essential in patients with rheumatoid arthritis (RA) as immunosuppressive medications increase their risk of invasive pneumococcal disease (IPD). However, data regarding the impact of timely administration of these vaccines on the risk of developing IPD are lacking for RA patients. We conducted a retrospective cohort study to assess the impact of on-time vaccination for pneumococcal conjugate vaccine (PCV) 13 and pneumococcal polysaccharide vaccine (PPSV) 23 in patients treated for RA on the development of IPD using national Veterans Affairs data from 2010 to 2018. Patients > 18 years of age, diagnosed with RA, and newly initiated on RA treatment were included. Pneumococcal vaccine compliance was assessed by measuring on-time receipt of PCV13 and PPSV23 vaccinations. A total of 33,545 patients were included in the cohort. Non-compliance with PCV recommendations was associated with an increased risk of IPD in a multivariable logistic regression model. This finding was consistent whether IPD status was ascertained by International Classification of Diseases coding (OR 2.42, 95%CI 2.14–2.73) or microbiologic data (OR 1.64, 95%CI 1.26–2.14). Providers should actively seek opportunities to provide pneumococcal vaccinations to patients with RA, as their on-time administration is associated with a decreased risk of IPD.

## 1. Introduction

Invasive pneumococcal disease (IPD) is a serious infection that can lead to death. It is one of the top ten causes of death in the United States [1]. Multivalent pneumococcal vaccines have been introduced around the world, and they have significantly reduced the burden of disease caused by *Streptococcus pneumoniae* infection [2,3]. Patients with immunocompromising conditions, such as rheumatoid arthritis (RA), face an elevated risk of invasive pneumococcal disease (IPD) due to the immunosuppressive medications commonly prescribed to manage their condition, such as disease-modifying anti-rheumatic drugs (DMARDs). DMARDs are categorized into three groups: conventional, biologic, and targeted [4]. Conventional synthetic DMARDs broadly suppress the overactive immune system, while biologic DMARDs inhibit specific targets like tumor necrosis factor alpha or interleukins. Targeted synthetic DMARDs act by inhibiting a wider range of cytokines. Previous data have shown that patients with RA who develop pneumonia are at an increased risk of death [5]. Immunization is a key strategy for this group to prevent infections. In adults <65 years of age with underlying conditions like treated RA, pneumococcal conjugate vaccine (PCV) 13 serotypes accounted for 30% of IPD; serotypes unique to PCV15, PCV20, and pneumococcal polysaccharide vaccine (PPSV) 23 caused 13%, 28%, and 43% of IPD, respectively [6]. However, reports of increases in non-PCV13 strains associated with antibiotic resistance could decrease the expected benefit of PCV13 administration [7]. Until 2022, patients receiving immunosuppressant medications for RA were advised to receive two different pneumococcal vaccines, PCV13 and PPSV23, to protect them against the most common strains of *S. pneumoniae* [8].

Vaccine uptake among RA patients is also suboptimal with vaccination rates for pneumococcus reported to be ~42% [9,10]. Previous studies evaluating compliance with multidose vaccine schedules, such as those for pneumococcus, are also suboptimal [11,12]. These findings are in agreement with an evaluation of potential barriers to on-time vaccination in patients with RA [13]. While vaccine uptake in this population continues to be a challenge, what is unknown is whether pneumococcus vaccine compliance is associated with IPD in patients on immunocompromising medications for RA. Therefore, the relative importance of providing pneumococcal vaccines to patients is unknown and may result in clinicians deprioritizing pneumococcal vaccinations compared to other interventions. Pneumococcal vaccination rates for patients with RA would hopefully increase if clinicians were armed with data to show that IPD rates are lowered in patients who receive this intervention.

The present study aimed to determine if pneumococcus vaccine compliance with both the PCV13 and PPSV23 vaccines was associated with lower IPD events. Therefore, we conducted a retrospective cohort study evaluating the on-time compliance for the PPSV23 and PCV13 vaccines and IPD in patients receiving immunosuppressive medications for RA in the Veterans Affairs (VA) Healthcare System.

## 2. Materials and Methods

### 2.1. Study Design

We performed a retrospective cohort study to assess the impact of on-schedule PCV13 and PPSV23 vaccination compliance on IPD outcomes in patients with treated RA between 2010 and 2018. Patient data were retrieved from the VA Corporate Data Warehouse, encompassing information from 1293 healthcare facilities, including 171 VA medical centers and 1112 outpatient sites. The Corporate Data Warehouse data include inpatient and outpatient diagnosis/procedure codes, pharmacy records, and laboratory results. This study was approved by the VA North Texas Health Care System (Institutional Review Board # 1589849-3) and Texas Tech University Health Sciences Center (Institutional Review Board # A20-4131). A waiver of informed consent was granted for this study.

### 2.2. Participants

All participants were adults (18 years of age or older) with RA who were being treated at a VA medical center and who had a new prescription for any immunosuppressant medication. To identify participants who received regular VA care, participants were included in the study if they had at least two visits to a VA clinic or one clinic visit and filled a prescription for any medication at a VA pharmacy within the 24 months prior to the index date defined below [14]. Patients with RA were identified using a validated algorithm that uses both inpatient and outpatient International Classification of Diseases (ICD)-10 (M06.9) and ICD-9 codes [15]. Patients were eligible and included in the study on the date of diagnosis with RA and initiation of immunosuppressant medication. The initiation date, defined as the release of the medication or clinic administration to the patient of immunosuppressing medications, was considered the index date. Immunosuppressing medications included methotrexate, leflunomide, adalimumab, certolizumab pegol, etanercept, golimumab, infliximab, abatacept, rituximab, tocilizumab, and tofacitinib. We did not include steroid medications since they can be given for multiple indications besides RA and are usually given in short courses. To ensure new RA immunosuppressant users, we excluded patients with prior prescriptions for immunosuppressants during a 2-year look-back period prior to the first identified immunosuppressant therapy.

### 2.3. Outcome

The outcome of interest was IPD. This was defined as an inpatient discharge diagnosis for pneumococcal meningitis (G00.1), sepsis (A40.3), pneumonia (J13), infection NOS (B95.3), or bacteremia (R78.81) by ICD-10 codes after May 2015 [16]. The ICD-9 codes considered for this outcome prior to May 2015 are in Appendix A. All identifying ICD-9 and ICD-10 codes had to be in either the primary or secondary reason for admission to the hospital. This was denoted as outcome 1. For greater precision, we evaluated another outcome, labeled outcome 2. This was defined as an inpatient discharge diagnosis of pneumococcal septicemia, pneumococcus infection in conditions classified elsewhere, and pneumococcal meningitis by ICD-9 codes. Outcome 2 was further defined by ICD-10 codes sepsis due to *S. pneumoniae*, bacteremia, pneumococcal meningitis, and *S. pneumoniae* as the cause of diseases classified elsewhere. Other parameters that defined outcome 2 were having ICD-9 codes listed in Appendix A and having microbiological cultures positive for *S. pneumoniae*. To ensure that these events were incident events, we looked back one year prior to the index date and excluded any patients who had a previous event. Given the definition of exposures below, we included a lag period of 1 year after the index date. Any patients who had events within this one-year lag period were excluded from the analyses. This was implemented to avoid immortal time bias that would occur if we started follow-up at index, especially for those that received only one of the two vaccinations [17].

### 2.4. Exposures

Vaccine compliance was the primary exposure of interest, defined as the timely administration of PCV13 and PPSV23 vaccinations. Vaccine administration was determined using pharmacy fill records or clinic administration data. For patients who were pneumococcal vaccine-naïve, PCV13 should be administered first, followed by PPSV23 at least 8 weeks later. Patients who received PPSV23 within 8 weeks of their PCV13 dose were classified as non-compliant. For those not pneumococcal vaccine-naïve and who received PPSV23 prior to follow-up, a PCV13 dose should be given at least 1 year after the last PPSV23 dose. Receiving PCV13 within 1 year of a PPSV23 dose was also considered as non-compliance. These definitions were based on recommendations from the Advisory Committee on Immunization Practices at the United States Centers for Disease Control and Prevention [8,18].

### 2.5. Covariates

Covariates were collected and defined within the year prior to the start of follow-up. Patient-level factors, including age, sex, race/ethnicity, marital status, and VA priority status (serving as a proxy for needs-based or disability care), were assessed. Veterans are assigned to a priority group based on their military service history, disability rating, income level, Medicaid eligibility, or receipt of VA pension benefits. Veterans with service-connected disabilities are given the highest priority (priority status < 3), while those with higher incomes and no service-connected disabilities are assigned a lower priority. Priority status can impact how much (if anything) a veteran will have to pay toward the cost of care. Patient comorbidities were captured as summarized by the Charlson–Deyo comorbidity index. Specifically extracted conditions included previous myocardial infarction, peripheral vascular disease, cerebrovascular disease, dementia, chronic pulmonary disease, peptic ulcer disease, mild liver disease, hemiplegia or paraplegia, renal disease, malignancy (excluding malignant neoplasm of the skin), moderate or severe liver disease, metastatic solid tumor, and human immunodeficiency virus/acquired immunodeficiency syndrome [19]. We also evaluated RA treatment as listed above.

### 2.6. Statistical Methods

Patient- and system-level characteristics were summarized using descriptive statistics. The proportion of patients that were considered compliant was calculated by dividing the number of compliant patients by the number of patients in the overall cohort. Continuous variables were described using mean and standard deviations, and categorical variables were reported as a number of unique subjects and the proportion of patients in either the compliant or non-compliant group.

The crude incidence rate of IPD events in the full study cohort was calculated with 95% confidence intervals (95% CIs) based on the Poisson distribution.

Cox proportional hazards models were constructed to calculate hazard ratios (HRs) and 95% CIs comparing vaccine compliance on IPD events. The proportional hazard assumption was examined by checking Schoenfeld residuals over time [20]. Robust standard errors were used when calculating 95% CI to improve the reliability and validity of the statistical inference. Kaplan–Meier curves were constructed for each cohort described. Statistical analyses were conducted using SAS, version 9.4 (SAS Institute), and R, version 4.4.1.

## 3. Results

Baseline characteristics of the 33,545 patients included in the cohort are shown by compliance status in Table 1. A total of 80,482 patient-years were included in the analyses. The compliance rate with the recommended PCV vaccination schedule was 42%. Of those that were vaccine non-compliant, 10,498 did not receive any pneumococcal vaccination, 6499 received PCV13 only, 2168 received PPSV23 only, and 293 received both vaccinations but not on schedule. The average age at the index date was 61.1 (12.8), with a majority of the cohort being white (n = 25,176, 75%) males (n = 28,293, 84%). Patients were mostly married (n = 19,403, 58%), non-Hispanic (n = 30,538, 91%), and had a priority status of 3 or greater (n = 18,802, 56%). Patients were mostly prescribed methotrexate (n = 13,950) and hydroxychloroquine (n = 12,155) for RA. Biologics were prescribed in 3211 patients for RA. Adalimumab (n = 1607) and etanercept (n = 13,284) were the most commonly prescribed biologics. Sulfasalazine, tofacitinib, and leflunomide were also prescribed (n = 4229).

In the total cohort, 1056 patients experienced outcome 1, and 225 experienced outcome 2. Of patients who were non-compliant, 3.4% (n = 652) experienced outcome 1, and 0.59% (n = 115) experienced outcome 2. The crude IPD outcome rate (outcome 1) was 0.013 events per patient-year (95% CI 0.012–0.014). The crude IPD rate for outcome 2 was 0.003 events per patient-year (95% CI 0.002–0.004).

In a multivariable logistic regression model accounting for other potential confounding variables, non-compliance with PCV recommendations was associated with an increased risk of IPD. This finding was consistent whether ascertained by ICD coding (HR 2.42, 95% CI 2.14–2.73) or microbiologic data for IPD (HR 1.64, 95% CI 1.26–2.14). Figure 1 displays the Kaplan–Meier curves for outcome 1, and Figure 2 displays the Kaplan–Meier curves for outcome 2.

## 4. Discussion

Non-compliance with PCV recommendations was associated with an increased risk of IPD in this nationwide cohort of veteran patients with treated RA. This increased risk was found regardless of whether the presence of IPD was determined through ICD codes or microbiologic data. This is clinically meaningful as IPD causes more severe infections than non-IPD due to *S. pneumoniae* and is a top 10 cause of death [1].

The beneficial impact of pneumococcal vaccinations on IPD rates is clear in immunocompetent populations [21]. The introduction of pneumococcal vaccines has significantly reduced IPD in immunocompromised patients. Research indicates that PCV13 has decreased the overall incidence of IPD by 40.3% and reduced PCV13-type IPD by 72.5% in individuals with human immunodeficiency virus, demonstrating its effectiveness in this high-risk, immunocompromised group [22]. Furthermore, the indirect benefits of pediatric PCV13 use have lowered IPD rates among adults with immunocompromising conditions [23]. The impact of the pneumococcal vaccine on IPD in RA patients receiving immunosuppressant medications is significant. Research indicates that pneumococcal vaccine uptake is low among patients with inflammatory rheumatic diseases, particularly in younger individuals and those not on biological therapy [13,24]. Two studies have also shown this benefit in patients with either immunocompromising conditions or those receiving immunosuppressive medications [25,26]. Our study strengthens the evidence supporting pneumococcal vaccine schedule compliance specific to patients with RA, as the previous study only included 41 patients [25]. The earlier study that included patients with RA was limited to Toronto and the Peel region of Canada, unlike our analysis, which included all veterans in the United States. The paucity of outcome data supporting the impact of pneumococcal vaccinations in adults treated for RA underscores the lack of attention the increased risk of IPD receives compared to their predisposing condition. A recent meta-analysis evaluated seroconversion rates in patients receiving various RA treatments versus healthy controls but did not evaluate clinical outcomes [27]. However, the meta-analysis did not use the one study that used both PPSV23 and PCV13 that met the screening criteria [28]. That study concluded that the vaccine response in patients receiving immunosuppressive therapy was decreased compared to the absence of such therapy. However, only 14% (n = 30) of the study population had RA as an underlying disease and no vaccine response data specific to the RA subgroup were presented. A separate study that was not presented by the meta-analysis suggests that opsonophagocytosis killing assay (OPKA) titers were similar for most DMARDs in 182 patients (RA = 63 patients) with autoimmune inflammatory diseases [29]. The exceptions were for patients on etanercept to have OPKA titers against more serotypes and adalimumab to have fewer compared to other biological treatments. However, neither study evaluated clinical outcomes related to the incidence of IPD in these patients.

This study provides primary care providers, rheumatologists, nurses, and pharmacists involved in caring for patients with RA further evidence that pneumococcal vaccination impacts clinically meaningful outcomes in this population. Various strategies have been explored to enhance pneumococcal vaccine uptake in patients with RA. Research has shown that interventions aimed at patients, physicians, or addressing barriers have successfully increased vaccination rates in individuals with immune-mediated conditions like RA, with barrier-focused interventions proving to have the most significant positive impact [30]. Nurse-based interventions have been studied with modest increases in pneumococcal vaccine uptake [31]. Interventions targeting provider prescribing habits have also been implemented, yielding modest impact [32]. Although the overall IPD rates were low, the difference between on-time vaccination was detected even when using fewer IPD events with microbiologic documentation. Given that some patients with RA may look otherwise healthy, it may be easy to overlook or ignore a best-practice alert to ensure optimal vaccination status [33]. The vaccination compliance rate of ~42% in this study is similar to what others have reported [9,10]. This highlights the glaring need for improvements in compliance with pneumococcal vaccination schedule adherence in this population and the opportunity to improve the health of these patients further.

We acknowledge that our study has several limitations, including its observational approach. On the other hand, this study design offers the advantage of observing and evaluating the real-world practices of pneumococcal vaccination among rheumatoid patients, as well as the approaches of their healthcare providers and systems in administering and timing these vaccinations. Observational studies cannot fully account for unmeasured confounding, but our findings were consistent when analyzing IPD rates by ICD codes or microbiologic data. Additionally, we adjusted for many of the measured confounders that could influence the association between pneumococcal vaccination and IPD outcomes. We were unable to evaluate the impact of PCV20 on IPD as the number of patients would be less than our cohort, given how recent the change in practice was compared to the time period of our study. The majority of patients in our study sample were white males which may limit the generalizability to a more diverse population, particularly since RA affects women. However, the large number of patients, including women, gives us confidence in the findings of this study. Additionally, biomarkers such as anti-citrulline antibodies, rheumatoid factor, C-reactive protein, or vaccine antibody titers were not collected for the cohort as this was not the focus of the current study. Lastly, the choice was made to end the cohort time period at 2018 to avoid potential confounding from the COVID-19 pandemic.

Future research in this area could focus on some of these items including the association between inflammatory biomarkers and IPD incidence. Similarly, the impact of the type of DMARD on the incidence of IPD could be assessed. Educational interventions using the results of this study to encourage pneumococcal vaccination in patients with RA could be evaluated prospectively. The impact of PCV20 on IPD rates will be more easily evaluated once more years of its use are available to create a larger cohort of patients. Additionally, the impact of pneumococcal vaccines on IPD incidence in patients receiving immunosuppressive medications for other diseases could be evaluated.

## 5. Conclusions

In this study, utilizing a national cohort of veterans, non-compliance with PCV15 recommendations for patients with RA was associated with increased rates of IPD. IPD is a top 10 cause of death and represents more severe infections than non-IPD cases. Providers and systems should use these results to encourage compliance with pneumococcal vaccination recommendations in patients treated for RA.

## Figures and Tables

**Figure 1 microorganisms-12-02073-f001:**
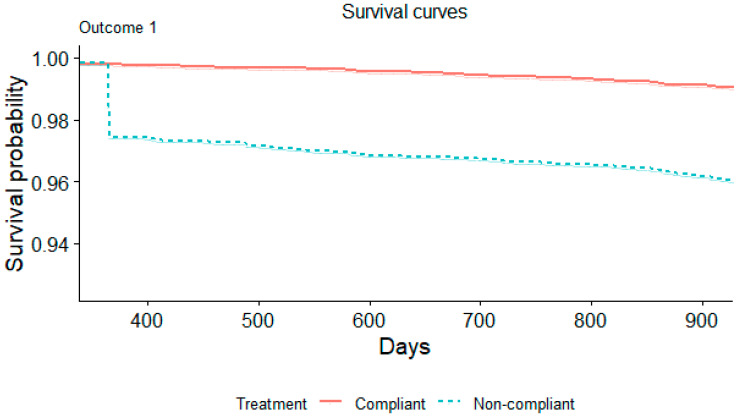
Kaplan–Meier survival curve illustrating the proportion of patients experiencing outcome 1 based on their vaccine compliance status.

**Figure 2 microorganisms-12-02073-f002:**
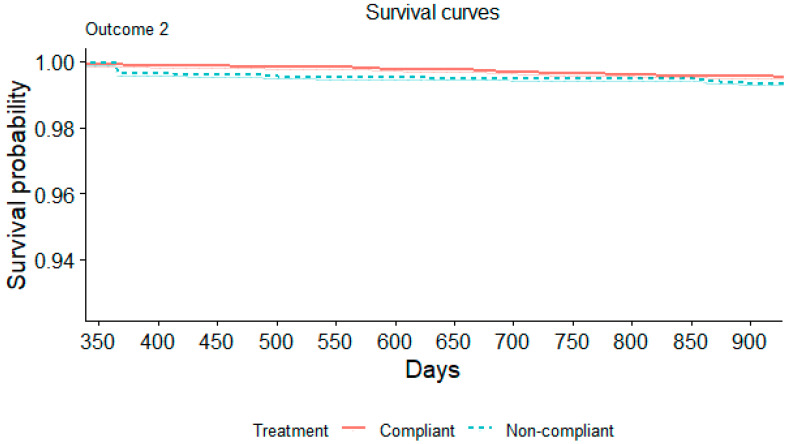
Kaplan–Meier survival curve illustrating the proportion of patients experiencing outcome 2 based on their vaccine compliance status.

**Table 1 microorganisms-12-02073-t001:** Baseline characteristics of the cohort (n (%)).

	Compliant (n = 14,087)	Non-Compliant (n = 19,458)
Age at diagnosis (mean, sd)	61.8 (10.4)	60.0 (14.3)
Age at treatment (mean, sd)	62.1 (10.4)	60.3 (14.3)
Race		
American Indian/Alaska Native	162 (1.1)	230 (1.2)
Asian	115 (0.8)	129 (0.7)
Black or African American	2033 (14.4)	3487 (17.9)
Missing/Unknown	618 (4.4)	1229 (6.3)
Multi-Race	30 (0.2)	38 (0.2)
Native Hawaiian/PI	132 (0.9)	166 (0.9)
White	10,997 (78.1)	14,179 (72.9)
Ethnicity		
Not Hispanic or Latino	12,893 (91.5)	17,645 (90.7)
Hispanic or Latino	800 (5.7)	1123 (5.8)
Refused to Answer	189 (1.3)	293 (1.5)
Unknown/Missing	205 (1.4)	397 (2.1)
Male Sex	12,270 (87.1%)	16,023 (82.3)
Marital Status		
Divorced	3301 (23.4)	4707 (24.2)
Married	8455 (60.0)	10,948 (56.3)
Separated	424 (3.0)	653 (3.4)
Single	1139 (8.1)	1916 (9.8)
Unknown/Missing	53 (0.4)	104 (0.5)
Widowed	715 (5.1)	1130 (5.8)
Priority Status		
1	6319 (44.9)	8005 (41.1)
2	147 (1.0)	272 (1.4)
3	2655 (18.8)	3596 (18.5)
4	252 (1.8)	330 (1.7)
5	4714 (33.5)	7255 (37.3)
Treatment		
Abatacept	37 (0.3)	43 (0.2)
Adalimumab	724 (5.1)	883 (4.5)
Anakinra	18 (0.1)	32 (0.2)
Certolizumab	25 (0.2)	40 (0.2)
Etanercept	627 (4.5)	701 (3.6)
Golimumab	8 (0.1)	12 (0.1)
Hydroxychloroquine	4676 (33.2)	7479 (38.4)
Infliximab	15 (0.1)	11 (0.1)
Leflunomide	523 (3.7)	641 (3.3)
Methotrexate	6169 (43.8)	7781 (40.0)
Rituximab	2 (0.0)	2 (0.0)
Sulfasalazine	1233 (8.8)	1780 (9.1)
Tocilizumab	15 (0.1)	16 (0.1)
Tofacitinib	15 (0.1)	37 (0.2)
Drug Class		
csDMARD	10,845 (77.0)	15,260 (78.4)
bDMARD	1471 (10.5)	1740 (9.0)
tsDMARD	1771 (12.6)	2458 (12.6)
Comorbidities		
Myocardial infarction	260 (1.8)	299 (1.5)
Congestive heart failure	618 (4.4)	850 (4.4)
Peripheral vascular disease	803 (5.7)	1037 (5.3)
Cerebrovascular diseases	607 (4.3)	821 (4.2)
Dementia	61 (0.4)	115 (0.6)
COPD	2818 (20.0)	3443 (17.7)
Peptic ulcer disease	161 (1.1)	212 (1.1)
Mild liver disease	609 (4.3)	742 (3.8)
Diabetes without comp	3617 (25.7)	3844 (19.8)
Diabetes with comp	1053 (7.5)	963 (4.9)
Hemi/Paraplegia	71 (0.5)	85 (0.4)
Renal disease	729 (5.2)	908 (4.7)
Any malignancy	1033 (7.3)	1322 (6.8)
Moderate or severe liver disease	22 (0.2)	32 (0.2)
Metastatic solid tumor	37 (0.3)	68 (0.3)
AIDS/HIV	45 (0.3)	25 (0.1)

Abbreviations: csDMARD = conventional disease-modifying anti-rheumatic drug, bDMARD = biologic disease-modifying anti-rheumatic drug, tsDMARD = targeted disease-modifying anti-rheumatic drug, AIDS = acquired immunodeficiency syndrome; comp = complications; COPD = chronic obstructive pulmonary disease; HIV = human immunodeficiency virus; n = number; PI = Pacific Islander; SD = standard deviation.

## Data Availability

The authors do not have permission to share data, per Veteran Affairs policy. The authors will share code to extract and analyze data if contacted.

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
