# Peer review of "Impact of PCV13 and PPSV23 Vaccination on Invasive Pneumococcal Disease in Adults with Treated Rheumatoid Arthritis: A Population-Based Study"

_microorganisms, 2024, doi:10.3390/microorganisms12102073_

Round 1
Reviewer 1 Report
Comments and Suggestions for Authors
I congratulate the authors, the work is very interesting, well structured and well written and scientifically valid. I have only a few minor remarks.
Minor questions:
1) In the introduction section, the authors briefly describe the purpose of this scientific study (lines 52 - 56). The authors evaluated compliance for PPSV23 and PCV13 vaccines in patients with rheumatoid arthritis undergoing immunosuppressive therapy. I would suggest that the authors include a very brief description (no more than two lines) of the type of immunosuppressive drugs used and their characteristics. For example, methotrexate, a folic acid synthesis antagonist, rituximab, a monoclonal antibody against CD20 expressed on B lymphocytes, abatacept, a monoclonal chimeric antibody analog of CTLA-4 that acts as an inhibitor of T lymphocyte activation, etc..
2) In the materials and methods section the authors describe the study setup (lines 58 - 68). I would suggest the authors to insert a very brief summary comment indicating the total number or overall number of subjects involved in the study and their average age, with, if known, the average age of onset of rheumatoid arthritis. (if the data is known).
3) In the materials and methods section the authors briefly describe the characteristics of the patients affected by rheumatoid arthritis studied (lines 69 - 87). It would be interesting to have a reference to some laboratory data that define the status of a subject affected by rheumatoid arthritis. For example, if the positivity and/or levels of anti-citrulline antibodies (anti-CCP) or rheumatoid factor and C-reactive protein (CRP) at the onset and/or diagnosis of rheumatoid arthritis are known. If such data are known, they should be reported in a table (perhaps as an average value) for the different categories considered. This would embellish and enrich the work.
4) In the materials and methods section (lines 108 - 119) the authors describe the vaccination methods and the related categories, in the patients studied. In my opinion it would be very useful to know the antibody titers generated after vaccination. If the levels of anti-pneumococcus antibodies (IgG) were measured (for example after 2 months or six months) at the end of the vaccination cycle, such data should be shown and commented. And this should also be done in relation to the different immunosuppressive treatments to which the patients were subjected. This is to highlight whether, possibly, the use of different immunosuppressive drugs affects the effectiveness of the response to the vaccine. If such analysis has been carried out, it should be included to make the work even stronger and more interesting.
Author Response
Reviewer 1
I congratulate the authors, the work is very interesting, well structured and well written and scientifically valid. I have only a few minor remarks.
Thank you for your positive feedback and for recognizing the value and structure of our work. We appreciate your kind words and are grateful for the constructive remarks you provided. We have carefully considered your suggestions and addressed them to further strengthen the manuscript.
Minor questions:
1) I would suggest that the authors include a very brief description (no more than two lines) of the type of immunosuppressive drugs used and their characteristics. For example, methotrexate, a folic acid synthesis antagonist, rituximab, a monoclonal antibody against CD20 expressed on B lymphocytes, abatacept, a monoclonal chimeric antibody analog of CTLA-4 that acts as an inhibitor of T lymphocyte activation, etc.
We appreciate your suggestion regarding the inclusion of the mechanisms of action for each drug. While we felt that a detailed description of each drug’s mechanism might be outside the scope of this manuscript, as it is not intended to serve as a review article, we have added the following information to address the immunosuppressive impact of each type of DMARD:
'However, patients with underlying immunocompromising conditions, such as rheumatoid arthritis (RA), are at an increased risk of IPD due to the immunosuppressive medications commonly used in treatment, such as disease-modifying anti-rheumatic drugs (DMARDs). DMARDs can be classified as conventional, biologic, or targeted [4]. Conventional synthetic DMARDs result in broad immunosuppression. Biologic DMARDs target specific molecules like tumor necrosis factor alpha or interleukins, while targeted synthetic DMARDs inhibit a range of cytokines.'
2) In the materials and methods section the authors describe the study setup (lines 58 - 68). I would suggest the authors to insert a very brief summary comment indicating the total number or overall number of subjects involved in the study and their average age, with, if known, the average age of onset of rheumatoid arthritis. (if the data is known).
Thank you for your suggestion. The information you requested is presented in Table 1. We believe it is most appropriate to keep this data within the Results section, as per standard practice, but we appreciate your input and consideration.
3) In the materials and methods section the authors briefly describe the characteristics of the patients affected by rheumatoid arthritis studied (lines 69 - 87). It would be interesting to have a reference to some laboratory data that define the status of a subject affected by rheumatoid arthritis. For example, if the positivity and/or levels of anti-citrulline antibodies (anti-CCP) or rheumatoid factor and C-reactive protein (CRP) at the onset and/or diagnosis of rheumatoid arthritis are known. If such data are known, they should be reported in a table (perhaps as an average value) for the different categories considered. This would embellish and enrich the work.
Thank you for the suggestion. However, we did not collect data regarding these laboratory markers as it was not the focus of the current study. Therefore, we have added the following sentence to the limitations section of the discussion to acknowledge this:
“Additionally, the collection of biomarkers such as anti-citrulline antibodies, rheumatoid factor, C-reactive protein, or vaccine antibody titers were not collected for the cohort as this was not the focus of the current study.”
4) In the materials and methods section (lines 108 - 119) the authors describe the vaccination methods and the related categories, in the patients studied. In my opinion it would be very useful to know the antibody titers generated after vaccination. If the levels of anti-pneumococcus antibodies (IgG) were measured (for example after 2 months or six months) at the end of the vaccination cycle, such data should be shown and commented. And this should also be done in relation to the different immunosuppressive treatments to which the patients were subjected. This is to highlight whether, possibly, the use of different immunosuppressive drugs affects the effectiveness of the response to the vaccine. If such analysis has been carried out, it should be included to make the work even stronger and more interesting.
Thank you for your valuable comment. The collection of antibody titers following vaccination is not part of the standard care for RA patients in the VA, so unfortunately, these data are not available. To address this, we have added the following sentence to the limitations section:
'Additionally, the collection of biomarkers such as anti-citrulline antibodies, rheumatoid factor, C-reactive protein, or vaccine antibody titers is not standard practice in VA rheumatology clinics. As a result, these tests were not included in our data analysis.'
Reviewer 2 Report
Comments and Suggestions for Authors
The manuscript by Alvarez et al evaluates the importance of compliance of vaccination in patients with rheumatoid arthritis. However, I have some comments that should be addressed by the authors:
1. In the abstract rheumatoid arthritis is abbreviated as RA but later during the rest of the manuscript this abbreviation was not used any more. Any explanation? I suggest that if you use this abbreviation, you should use it every time rheumatoid arthritis is mentioned. Otherwise, delete it from the abstract
2. Streptococcus pneumoniae (whole name of the bacterium) is mentioned in page 1 line 3. For further mentions of the microorganism in the text do not use the entire name. Use S. pneumoniae instead. (page 1 line 43, page 4 lines 98 and 101)
3. Cites 2 and 3 in paragraph of page 1 lines 31-33 are very old (2008 and 2011) to explain that PCVs have reduced the burden of disease. I will cite more recent studies and even I will mention that PCVs have decline serotypes associated with antibiotic resistance and I will mention that serotype replacement is a serious threat after the use of PCVs. In that paragraph I will cite the following references covering all these aspects (PMID: 38906265; PMID: 35932764; PMID: 38310905; PMID: 39116949)
4. The name S. pneumoniae is repeated twice in line 98 page 4. Please, delete one
5. Page 7 lines 212-214: There is a study evaluating the immune response of vaccination with PCV13 plus PPSV23 in patients with different rheumatological diseases including rheumatoid arthritis and treated with different biological drugs that should be mentioned here or maybe discuss this manuscript in a new paragraph. It is PMID: 33671007 (Richi P et al, Vaccines (Basel). 2021 Feb 28;9(3):203. doi: 10.3390/vaccines9030203)
Author Response
- In the abstract rheumatoid arthritis is abbreviated as RA but later during the rest of the manuscript this abbreviation was not used any more. Any explanation?
Thank you for noting the inconsistency. We will add the abbreviation to the manuscript to be consistent. As to the reason for the inconsistency, we used the abbreviation in the abstract due to the word count limitation. We did not abbreviate previously in the main text to increase the word count.
- Streptococcus pneumoniae (whole name of the bacterium) is mentioned in page 1 line 3. For further mentions of the microorganism in the text do not use the entire name. Use S. pneumoniae instead. (page 1 line 43, page 4 lines 98 and 101)
Thank you for noting this issue and we have now addressed it throughout the manuscript.
- Cites 2 and 3 in paragraph of page 1 lines 31-33 are very old (2008 and 2011) to explain that PCVs have reduced the burden of disease. I will cite more recent studies and even I will mention that PCVs have decline serotypes associated with antibiotic resistance and I will mention that serotype replacement is a serious threat after the use of PCVs. In that paragraph I will cite the following references covering all these aspects (PMID: 38906265; PMID: 35932764; PMID: 38310905; PMID: 39116949)
Thank you for your valuable critique. We have responded to each citation below:
PMID: 38906265 We are declining to add this reference as it primarily deals with the time frame of the COVID-19 pandemic. Our study timeframe was purposely chosen to avoid issues associated with COVID-19.
PMID: 35932764 We have added the following sentence to the introduction: “However, reports of increases in non-PCV13 strains associated with antibiotic resistance could decrease the expected benefit of PCV13 administration [7].”
PMID: 38310905 We are declining to add this reference as it is focused on infants and again deals with the timing and impact of the COVID-19 pandemic.
PMID: 39116949 We are declining to add this reference as it is focused on pediatrics and again deals with the timing of the COVID-19 pandemic.
- The name S. pneumoniae is repeated twice in line 98 page 4. Please, delete one
Thank you for noting this potential area of confusion. We have now moved “Streptococcus pneumoniae as the cause of diseases classified elsewhere” to the end of this list to separate the two S. pneumoniae mentions in the sentence to clarify that each reference is unique.
- Page 7 lines 212-214: There is a study evaluating the immune response of vaccination with PCV13 plus PPSV23 in patients with different rheumatological diseases including rheumatoid arthritis and treated with different biological drugs that should be mentioned here or maybe discuss this manuscript in a new paragraph. It is PMID: 33671007 (Richi P et al, Vaccines (Basel). 2021 Feb 28;9(3):203. doi: 10.3390/vaccines9030203)
We have added the requested reference and the following wording as context.
“However, the meta-analysis did not use the one study that used both PPSV23 and PCV13 that met the screening criteria [28]. That study concluded that the vaccine response in patients receiving immunosuppressive therapy was decreased compared to the absence of such therapy. However, only 14% (n = 30) of the study population had RA as an underlying disease and no vaccine response data specific to the RA subgroup was presented. A separate study that was not presented by the meta-analysis suggests that opsonophagocytosis killing assay (OPKA) titers were similar for most DMARDs in 182 patients (RA = 63 patients) with autoimmune inflammatory diseases [29]. The exceptions were for patients on etanercept to have OPKA titers against more serotypes and adalimumab to have fewer compared to other biological treatments. However, neither study evaluated clinical outcomes related to the incidence of IPD in these patients.”
Reviewer 3 Report
Comments and Suggestions for Authors
This article shows the implications of PCV13 and PPSV23 Vaccination on Invasive Pneumococcal Disease in Adults With Treated RA. The relevance of this manuscript to the scientific literature is currently unclear. The topic is relevant, but the major deficiencies identified in both content and form need to be addressed based on the specific recommendations below:
1. The template provided by the journal must be checked and filled in correctly and completely the affiliation and identification data.
2. It is crucial to explain why the 2010-2018 time frame was chosen given that we are in 2024 and what is the current relevance if the data is already quite old?
3. The conclusion part of the executive summary should be improved in terms of the results and what future research directions this research can refer to.
4. Abbreviations should be explained in the first sentence, then only used in abbreviated form (e.g. rheumatoid arthritis (RA) etc.), but if only mentioned once, the unabbreviated form should be used. Please review the basis on these principles throughout the manuscript, as the abstract is treated differently from the main text in this regard.
5. The introduction is much understated relative to the complexity of the topic. The approach related to AR needs to be expanded (briefly describing the pathology and its general management) and correlated with the increased susceptibility to infections in patients treated with bDMARDs in particular. I suggest you check it out and see: PMID: 34831081.
6. The scope of the paper should be improved in terms of the description of the contribution to the field analyzed and the elements of scientific novelty presented because the authors presented only in the last paragraph of the introduction what they did in the study.
7. Subsections 2.1-2.5 could be presented more clearly and concisely in the form of a figure/diagram showing the study design and the evaluator performed (algorithm), thus shortening the written text part.
8. Priority status should be explained in terms of implications.
9. L198-199 - the bibliographic index [x] is not placed after the period; correct throughout the entire manuscript.
10. Further detail the data in the conclusions and emphasize future research directions that address the current limitations of the study.
11. The bibliographic style is not the one used by the journal and should be updated to MDPI style.
Author Response
Reviewer 3
- The template provided by the journal must be checked and filled in correctly and completely the affiliation and identification data.
Thank you for this comment. We are unsure of what the author is referring to as we used the Microorganisms template on the journal's website and each author’s affiliations are denoted by the superscripts following each author’s name. If the editor wants us to provide each author’s email in the affiliations section we can do so. However, this was not required as part of our most recent publication in Microorganisms (https://www.mdpi.com/2076-2607/11/2/387).
- It is crucial to explain why the 2010-2018 time frame was chosen given that we are in 2024 and what is the current relevance if the data is already quite old?
Thank you for the opportunity to comment on the age of the dataset. The investigators chose to cut the cohort off at 2018 to avoid potential confounding associated with the COVID-19 pandemic. The following sentence was added to the limitations paragraph:
“Lastly, the choice was made to end the cohort time period at 2018 to avoid potential confounding from the COVID-19 pandemic.”
- The conclusion part of the executive summary should be improved in terms of the results and what future research directions this research can refer to.
Thank you for this comment. We believe the reviewer is referring to the abstract as the executive summary. If this is correct, the abstract word count limited our ability to expand to include more verbiage about our conclusions and/or future directions.
- Abbreviations should be explained in the first sentence, then only used in abbreviated form (e.g. rheumatoid arthritis (RA) etc.), but if only mentioned once, the unabbreviated form should be used. Please review the basis on these principles throughout the manuscript, as the abstract is treated differently from the main text in this regard.
Thank you for noting these issues. We have addressed them throughout the manuscript.
- The introduction is much understated relative to the complexity of the topic. The approach related to AR needs to be expanded (briefly describing the pathology and its general management) and correlated with the increased susceptibility to infections in patients treated with bDMARDs in particular. I suggest you check it out and see: PMID: 34831081.
Thank you for this comment. The wording below has been added to the introduction in an attempt to briefly address this comment. However, the current study did not aim to focus on the impact of DMARD type on the incidence of IPD or the relative protection of the PCV13/PPSV23 series against IPD based on DMARD type. These have been added to the discussion as potential future directions.
“Patients with immunocompromising conditions, such as rheumatoid arthritis (RA), face an elevated risk of invasive pneumococcal disease (IPD) due to the immunosuppressive medications commonly prescribed to manage their condition, such as disease-modifying anti-rheumatic drugs (DMARDs). DMARDs are categorized into three groups: conventional, biologic, and targeted [4]. Conventional synthetic DMARDs broadly suppress the overactive immune system, while biologic DMARDs inhibit specific targets like tumor necrosis factor alpha or interleukins. Targeted synthetic DMARDs act by inhibiting a wider range of cytokines.”
- The scope of the paper should be improved in terms of the description of the contribution to the field analyzed and the elements of scientific novelty presented because the authors presented only in the last paragraph of the introduction what they did in the study.
Thank you for this critique. We have added the following to the introduction to better describe the study’s contribution to the field and its novelty:
“Therefore, the relative importance of providing pneumococcal vaccines to patients is unknown and may result in clinicians deprioritizing pneumococcal vaccinations compared to other interventions. Pneumococcal vaccination rates for patients with RA would hopefully increase if clinicians were armed with data to show that IPD rates are lowered in patients who receive this intervention.”
- Subsections 2.1-2.5 could be presented more clearly and concisely in the form of a figure/diagram showing the study design and the evaluator performed (algorithm), thus shortening the written text part.
Thank you for your stylistic suggestion. After careful consideration, we have decided not to include a figure or diagram, as we believe the methods used in the study are best conveyed through the text for clarity and completeness. Additionally, the editor encouraged us to maintain a higher word count rather than reducing the text.
- Priority status should be explained in terms of implications.
Thank you for this critique. The following sentence has been added to section 2.5 to explain the implication of priority status:
“Priority status can impact how much (if anything) a veteran will have to pay toward the cost of care.”
- L198-199 - the bibliographic index [x] is not placed after the period; correct throughout the entire manuscript.
Thank you for noting this inconsistency in adhering to the journal’s style. We have made the necessary corrections.
- Further detail the data in the conclusions and emphasize future research directions that address the current limitations of the study.
Thank you for this thoughtful critique. While it is not entirely clear what additional details the reviewer would like in the conclusions, we have added the following section at the end of the discussion, outlining potential future research directions to address the limitations of the current study.
“Future research in this area could focus on some of these items including the association between inflammatory biomarkers and IPD incidence. Similarly, the impact of the type of DMARD on the incidence of IPD could be assessed. Educational interventions using the results of this study to encourage pneumococcal vaccination in patients with RA could be evaluated prospectively. The impact of PCV20 on IPD rates will be more easily evaluated once more years of its use are available to create a larger cohort of patients. Additionally, the impact of pneumococcal vaccines on IPD incidence in patients receiving immunosuppressive medications for other diseases could be evaluated.”
- The bibliographic style is not the one used by the journal and should be updated to MDPI style.
Thank you for noting this. We have now downloaded the appropriate MDPI style file for EndNote and updated the citations and bibliography with the MDPI style.
Round 2
Reviewer 3 Report
Comments and Suggestions for Authors
The authors have improved the manuscript based on suggestions or explained why they didn't modify some parts.